# Comparison of Postoperative Recovery between Balanced and Total Intravenous Anesthesia in Patients Undergoing Off-Pump Coronary Artery Bypass (OPCAB) Surgery: A Prospective, Single-Blind Randomized Study

**DOI:** 10.3390/ijerph20032310

**Published:** 2023-01-28

**Authors:** Dongho Kang, Minji Kim, Hong-Beom Bae, Seonho Moon, Joungmin Kim

**Affiliations:** 1Department of Anesthesiology and Pain Medicine, Chonnam National University, Hwasun Hospital, Hwasun 58128, Chonnam, Republic of Korea; 2Department of Anesthesiology and Pain Medicine, Medical School, Chonnam National University, Gwangju 61469, Republic of Korea; 3Department of Anesthesiology and Pain Medicine, Chonnam National University Hospital, Gwangju 61469, Republic of Korea

**Keywords:** anesthesia recovery period, coronary artery bypass, balanced anesthesia, cardiac procedures, anesthesia, intravenous, sevoflurane, propofol

## Abstract

Recovery after anesthesia has a significant impact on a patient’s return to daily life. This study was performed to compare the postoperative quality of recovery according to the method of anesthesia administered among patients undergoing OPCAB using the Korean version of the Quality of Recovery-40 (QoR-40K) questionnaire. This single-blind, prospective study (trial number: KCT0004726) was performed using a population of 102 patients undergoing OPCAB under general anesthesia. The patients were randomly assigned to one of two groups using a computer-generated list: a total intravenous anesthesia group (Group T) and a balanced anesthesia group (Group B). The QoR-40K score was measured preoperatively and at 24 and 48 h after extubation. There was no significant difference in the QoR-40K scores between the groups at 24 and 48 h after extubation. In addition, there were no significant differences between groups with respect to any of the five dimensions of QoR-40K at 24 and 48 h after extubation. Finally, there were no differences in the postoperative opioid consumption, time to extubation, or length of hospital stay. In this study, there was no difference in the QoR-40K score at 24 h after extubation between Groups T and B. Therefore, both methods of anesthesia are suitable for use when performing OPCAB.

## 1. Introduction

Advances in anesthesia and surgical procedures have been accompanied by decreased rates of perioperative mortality and morbidity, and the focus of anesthesia management is now shifting to the quality of recovery after surgery [1]. The quality of recovery after anesthesia not only affects the patient’s subjective satisfaction but also their recovery from surgery, length of hospital stay, and medical costs [2,3]. Accordingly, several methods have been devised to quantify satisfaction with recovery. These a include numerical score, postoperative quality recovery score (PQRS), and the quality of postoperative recovery (QOR). The Quality of Recovery-40 (QoR-40) survey, which consists of 40 questions, is one of the most widely used tools for this purpose. Since its development in English, it has been translated and used in nine countries [4,5].

Coronary artery bypass grafting is a widely performed surgical method involving the bypassing of a partially or completely blocked coronary artery using an autologous artery or vein as a graft [6]. Coronary artery bypass surgery is performed using either balanced anesthesia, in which the patient inhales the anesthetic, or total intravenous anesthesia. Previous studies suggested that the use of inhaled anesthesia for cardiac surgery reduced patient mortality and cardiac enzyme elevation due to the myocardia-protective action of the drug [7,8]. However, other studies have produced inconclusive results concerning the benefits of inhaled anesthetics [9,10]. Among others, a recent large-scale multi-center RCT (MYRIAD) showed comparable outcomes between volatile and intravenous anesthetics [11].

Poor-quality recovery after cardiac surgery leads to a decline in quality of life during the 3 months following surgery [12]. However, there have been no comparative studies of recovery from anesthesia according to the anesthetic method applied. We hypothesized that the degree of recovery of patients undergoing cardiac surgery would differ depending on whether total intravenous anesthesia or balanced anesthesia was administered. This study was performed to compare the quality of recovery after anesthesia between anesthesia methods among patients undergoing OPCAB using the Korean version of QoR-40 (QoR-40K).

## 2. Materials and Methods

### 2.1. Methods

This prospective, randomized, single-blind, controlled study was performed between March 2020 and August 2021 in single tertiary care hospital. The study protocol was approved by the Institutional Review Board of Chonnam National University Hospital (Approval number: CNUH-2020-015), and the study was registered at the Clinical Research Information Service of the Korea National Institute of Health (trial number: KCT0004726). All participants provided written informed consent before enrollment in accordance with the principles of the Declaration of Helsinki. Patients aged 20–79 years, corresponding to American Society of Anesthesiologists (ASA) physical status I–IV, and who were undergoing off-pump coronary artery bypass graft surgery (OPCAB) were enrolled in the study. Patients who were taking any sedatives, opioids, or sleep aids and those with a history of allergic reactions to any of the study drugs were excluded. Patients with physical disabilities, those who were illiterate, those who underwent conversion to coronary artery bypass grafting (i.e., cardiopulmonary bypass (CPB)), those with severe left ventricular dysfunction (ejection fraction < 30%), those who suffered from major organ failure before surgery (creatinine ≥ 2 mg/dL, AST/ALT ≥ 3 times higher than normal before surgery and showing a progressive increase, and neurological abnormalities), or who required continuous mechanical ventilation assistance after surgery were also excluded.

Pre-anesthesia visits and evaluations were conducted on the day before, or day of, surgery to confirm the patient’s general condition, preoperative test results, and medications. In addition, demographic information, height, weight, comorbidities, EuroSCORE II, and the number of affected coronary vessels were extracted from the patients’ medical records. At this time, a blinded investigator used the QoR-40K questionnaire to obtain baseline values for all patients. The patients were divided into two groups according to anesthesia method, i.e., a total intravenous anesthesia group (Group T) and balanced anesthesia group (Group B). The group assignment process was performed based on a computer-generated list of random numbers by a researcher who did not participate in the administration of anesthesia. Researchers investigating the QOR-questionnaire were unaware of the assignment.

After entering the operating room, each patient was monitored by 5-lead electrocardiography, pulse oximetry, and noninvasive blood-pressure monitoring. Before inducing anesthesia, we implanted a 22 or 20 G catheter into the radial artery and began continuous arterial blood-pressure monitoring. After recording the baseline vital signs, 8 L/min of oxygen was applied via a face mask and sufentanil (0.3 μg/kg) was infused. 

Group T received target-controlled infusion of propofol (2% Fresofol^®^; Fresenius Kabi Korea Ltd., Seoul, Korea) with the target plasma concentration set to 1.5–2.0 μg/mL. Infusion was increased in increments of 0.5 μg/mL until reaching a target bispectral index (BIS) of 40–60. In Group B, a bolus (1–1.5 mg/kg) of propofol (1% Fresofol^®^; Fresenius Kabi Korea Ltd.) was administered slowly until the target BIS was reached. Sufentanil infusion was started at the beginning of anesthesia and was then infused continuously with a mean arterial pressure (MAP) of 70–90 mmHg in both groups. After loss of consciousness was confirmed, rocuronium (0.8–1.0 mg/kg) was administered intravenously to facilitate intubation. During maintenance of anesthesia, propofol or sevoflurane administration was adjusted according to the target BIS of 40–60. Fluids and vasopressors were used if necessary. In addition, capnography parameters, pulmonary artery pressure, transesophageal echocardiography parameters, and body temperature were monitored for all patients.

The surgery was performed via median sternotomy by the same surgical team (led by one surgeon). During surgery, body temperature was maintained within the normal range using an external heated mattress. Anesthetic management information, such as surgery time, anesthesia time, sufentanil dose, infused fluid volume, transfusion units, inotropic drug usage, and number of anastomotic vessels, was recorded. Blood pressure and heart rate (HR) were recorded at each time point after anesthesia induction, skin incision, sternotomy, anastomosis, protamine injection, and arrival at the intensive care unit (ICU) or general ward. After the operation, 0.075 mg of palonosetron was administered to all patients and they were transferred to the surgical ICU while maintaining mechanical ventilation. Extubation was performed by the attending physician if the patient met the criteria for extubation (responsive, negative inspiratory force > 20 mmHg, core temperature > 36.5 °C, arterial pH > 7.3, and absence of uncontrolled arrhythmia). Postoperative pain control was achieved using an intravenous patient-controlled analgesia (PCA) device (fentanyl; basal 6 μg/h; bolus 6 μg; lockout 10 min). Tramadol (50 mg) was administered intravenously as an additional analgesic if the patient complained of pain (numeric rating scale (NRS) score ≥ 4). Postoperative nausea and vomiting (PONV) scores were recorded (0: no PONV, 1: nausea, 2: retching, and 3: vomiting). Rescue antiemetics were injected in patients with PONV scores ≥ 3.

The primary outcome was the QoR-40K score 24 h after extubation. Secondary outcomes were the QoR-40K score 48 h after extubation; peri-operative vital signs (MAP and HR); total amounts of sufentanil, fluid, and blood products administered during surgery; use of inotropics; time to extubation after surgery; length of stay in the ICU; and length of stay in the hospital. Pain score (NRS), use of rescue analgesics, post-operative fentanyl consumption, post-operative nausea and vomiting (PONV), and use of rescue antiemetics were also measured as secondary outcomes at the time of extubation and 24 and 48 h thereafter. All researchers in this study were anesthesiologists.

The patient’s HR and blood pressure at the time of extubation and 24 and 48 h thereafter, pain at the surgical site, the amount of fentanyl used in the pain-control device, the use of additional analgesics, and the use of antiemetics were determined from the medical records. The number of transplanted vessels was confirmed based on the operation records. At 24 and 48 h after extubation, the questionnaires were completed by the participants themselves or by responding to what the researcher read to them.

### 2.2. Sample Size and Statistical Analysis

A difference of 10 points in the QoR-40K score is associated with a 15% improvement in the quality of recovery [13]. The sizes of the two groups in this study that were required to achieve a meaningful difference in the QoR-40K scores were calculated, and the results are as follows: 46 subjects per group were required to achieve a power of 80% with a type 1 error rate of 0.05. Considering the potential for dropouts, a total of 102 study subjects were recruited (51 in each group).

Data were analyzed using SPSS 23.0 (IBM Corp., Armonk, NY). The Kolmogorov–Smirnov test was performed to examine the normality of the data. Continuous variables with a normal distribution were compared using the two-sample independent t test, and continuous variables that were not normally distributed were compared using the Mann–Whitney test. Categorical data were compared using Fisher’s exact test. In all analyses, *p* < 0.05 was taken to indicate statistical significance.

## 3. Results

A total of 106 patients were assessed for eligibility, and 102 gave informed consent and were randomly assigned to the study groups. In Group T, four patients were excluded because of conversion to on-pump CABG (n = 1), reintubation after extubation (n = 1), or the application of extracorporeal membrane oxygenation (ECMO) (n = 2) after surgery. In Group B, three patients were excluded due to conversion to on-pump CABG (n = 2) and the application of ECMO after surgery (n = 1). Statistical analyses were performed with data from 48 patients in Group B and 47 in Group T (Figure 1). 

There were no significant differences in the demographic characteristics, comorbidities, or medications between the two groups (Table 1), or in the operation time, anesthesia time, amount of fluid given during surgery, use of inotropics, amount of blood transfusion, or duration of intubation (Table 2).

The QoR-40K scores, preoperatively and at 24 and 48 h after extubation, are presented in Table 3. The preoperative (baseline) QoR-40K score was not different between the two groups (*p* = 0.102). There was no significant difference in the QoR-40K score at 24 h after extubation between Groups B and T (163.85 ± 15.32 vs. 164.80 ± 20.82, *p* = 0.825). There were no significant differences between the two groups in the scores for the five dimensions of the QoR-40K at 24 h or 48 h after extubation (*p* = 0.285). In addition, there was no significant difference in terms of post-operative pain (Table 4) or PONV (Table 5) at 24 h and 48 h after extubation between the two groups.

The HR was significantly lower in Group T than Group B at the time of skin incision and sternotomy (Figure 2). The MAP was not significantly different between the two groups (Figure 3). There were no significant differences between Groups B and T in terms of the length of ICU stay (26.56 ± 13.80 vs. 22.79 ± 8.40 h, *p* = 0.112) or length of hospital stay (13.79 ± 4.59 vs. 12.57 ± 3.33 days, *p* = 0.143). There were no serious complications, including death, after surgery in either group.

## 4. Discussion

In the present study, there were no difference in the QoR-40K scores at 24 h after extubation according to the method of anesthesia administered to patients undergoing OPCAB surgery, which was the primary outcome. In addition, there were no significant differences between the two groups in terms of the QoR-40K score at 48 h after extubation and with respect to any of the five dimensions of the QoR-40K at 24 or 48 h after extubation.

Patient satisfaction after surgery is not only related to the absence of post-operative complications or the maintenance of stable vital signs but also to functional recovery, such as subjective symptoms, psychological satisfaction, and physical independence. Therefore, a multifaceted approach is required [14,15,16]. The QoR-40 questionnaire consists of a total of 40 items distributed according to five dimensions: emotional state, physical comfort, psychological support, physical independence, and pain. Each item is scored on a scale ranging from 1 to 5, and the total scores range from 40 (worst quality of recovery) to 200 (best quality of recovery) points [4]. In December 2018, a Korean translation of QoR-40, the QoR-40K, was released, and a report verifying its validity and reliability was published [17].

Several clinical studies showed differing results regarding the quality of recovery according to the method of anesthesia administered for various surgeries. In endoscopic sinus surgery, the QoR-40 score at 6 h after surgery was significantly higher in a total intravenous anesthesia (using propofol) group compared with a desflurane group. In thyroid surgery, total intravenous anesthesia using propofol improved the overall recovery quality score by reducing PONV [13,18]. However, in patients undergoing laparoscopic hysterectomy, there was no difference in the overall recovery after surgery between inhalation and intravenous anesthesia groups [19]. In addition, the results of a meta-analysis on noncardiac surgery studies showed that neither method of anesthesia affected the quality of recovery [2]. In our study, the QoR-40 score and PONV were measured at 24 and 48 h after extubation; however, there was no significant difference.

In this study, there was no difference in the QoR-40K scores between the two groups. In previous studies comparing QoR-40 scores between patients treated with general anesthesia using total anesthesia and inhalational anesthesia, the dimensions showing group differences were physical comfort, including PONV, and physical independence [13,20]. In the present study, however, the incidence of PONV was not different between the groups. Risk factors for PONV include female sex, a history of motion sickness or PONV, nonsmoker status, and the use of opioid analgesics after surgery [21]. A large proportion of the subjects in this study were male, and palonosetron (0.075 mg) was administered prophylactically at the end of surgery to all patients (to reduce the incidence of complications due to PONV). In addition, the center’s recovery protocol after cardiac surgery includes admission to the Intensive Care Unit (ICU) immediately after surgery and the weaning of mechanical ventilation after recovery of spontaneous breathing. Patients staying in the ICU were sedated using dexmedetomidine, with the goal of reaching values between -2 and -1 on the Richmond Agitation Sedation Scale. The average time to mechanical ventilation weaning was approximately 14 h in both of our groups, which overlapped with the typical timing of complaints of PONV [22]. These differences are thought to have contributed to the reduction in the incidence of PONV and to the lack of a difference between the two groups with respect to these dimensions (in contrast to previous studies).

Pain is an important factor in patient satisfaction after surgery. In a previous study, preoperative administration of pregabalin to patients undergoing OPCAB surgery reduced the postoperative analgesic dose and pain score, which improved recovery scores [1]. In this study, there were no significant differences between the two groups with respect to pain scores at the time of extubation or 24 and 48 h thereafter. The average NRS scores in Groups B and T were 5.5 and 6.0 (*p* = 0.186) at the time of extubation, 4.6 and 4.4, (*p* = 0.540) at 24 h after extubation, and 3.0 and 3.1 (*p* = 0.935) at 48 h after extubation, respectively. There were no differences in PCA dose or rescue analgesic administration between the two groups. All patients underwent surgery performed by one surgeon and the same surgical team, and there was no difference in the amount of sufentanil administered during surgery between the two groups. Furthermore, there were no differences between the two groups in terms of the pain and total QoR-40K scores.

There were no differences between the two groups regarding the secondary outcomes of this study, i.e., the MAP during and after surgery, the dose of sufentanil administered during surgery, the use of inotropics, the time to extubation after surgery, the length of stay in the ICU, or the length of stay in hospital. There were also no differences between the two groups regarding the rate of conversion to on-pump during surgery or ECMO after surgery. 

This study had several limitations. First, objective laboratory data such as myocardial enzyme levels could not be collected; thus, differences between the two groups could not be determined. Second, the study was conducted at a single institution and predominantly incorporated males (Group B, 89.6% males; Group T, 83.0% males). Therefore, it is difficult to generalize the results of this study. Finally, as we included patients with left ventricular function (ejection fraction) ≥ 30%, we could not confirm the results in high-risk patients with severe left ventricular failure.

## 5. Conclusions

In this study, there were no differences in the QoR-40K score at 24 h after extubation between the patients undergoing OPCAB surgery with total intravenous anesthesia and balanced anesthesia using sevoflurane. In addition, there were no differences between the two groups regarding hemodynamic parameters except heart rate during surgery, nor were there any differences concerning the length of stay in the ICU or hospital.

## Figures and Tables

**Figure 1 ijerph-20-02310-f001:**
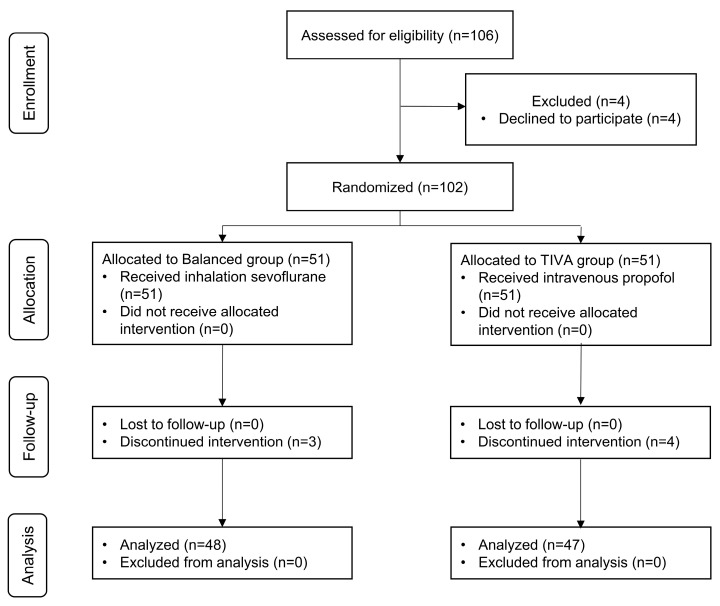
A flow chart that outlines patient selection, randomization, and analysis. TIVA, total intravenous anesthesia.

**Figure 2 ijerph-20-02310-f002:**
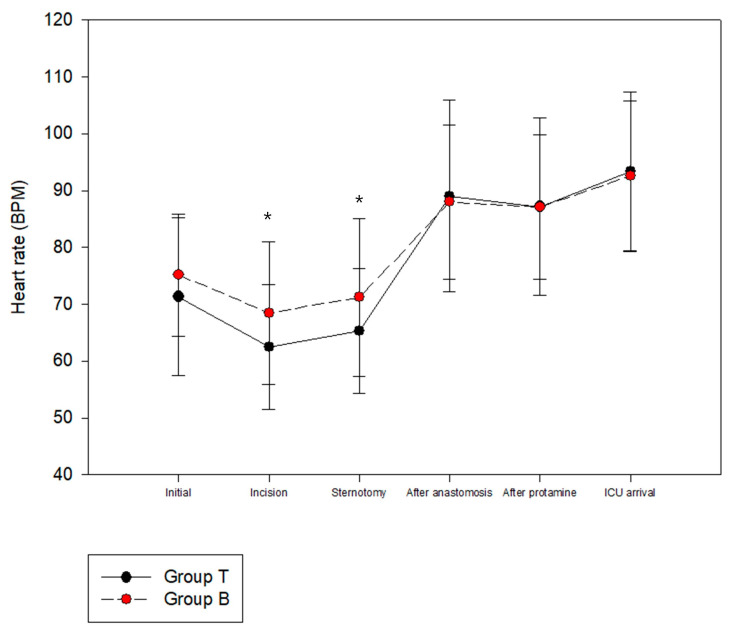
Perioperative hemodynamic change: Heart rate. * Represents statistical significance. (*p* < 0.05).

**Figure 3 ijerph-20-02310-f003:**
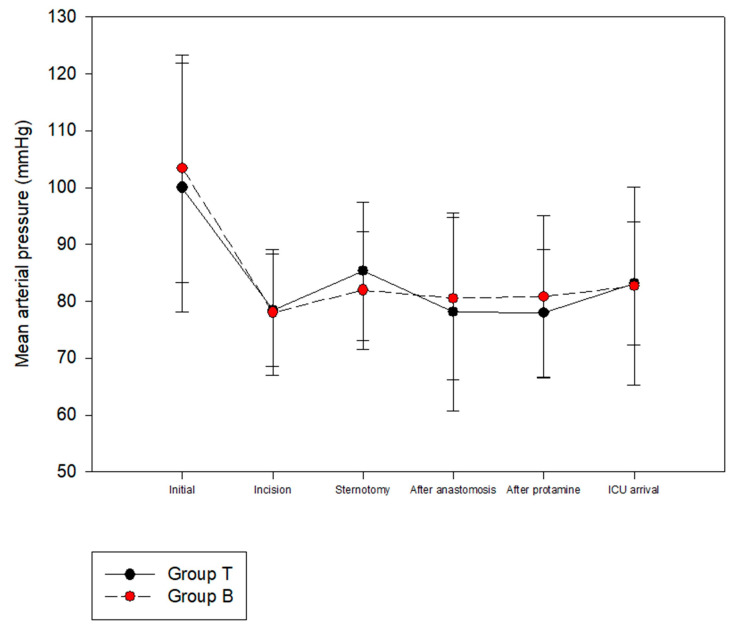
Perioperative hemodynamic change: Mean arterial pressure.

**Table 1 ijerph-20-02310-t001:** Patients’ baseline data.

		Group B (*n* = 48)	Group T (*n* = 47)	*p*-Value
Age		64.54 ± 9.36	66.04 ± 7.31	0.387
Gender (male, *n*) (%)		43 (89.6)	39 (83.0)	0.349
Weight (Kg)		68.38 ± 9.95	68.24 ± 12.25	0.950
Height (cm)		166.00 ± 7.40	164.20 ± 7.34	0.237
ASA-PS III/IV, *n* (%)		13/35 (27/73)	7/40 (15/85)	0.145
EuroSCORE (Ⅱ) (%)		1.33 ± 1.02	1.70 ± 1.00	0.081
Recent MI, *n* (%)		7 (15)	11 (23)	0.273
Comorbidity, *n* (%)	HTN	36 (75)	30 (64)	0.237
	DM	22 (46)	22 (47)	0.924
	COPD	1 (2)	2 (4)	0.545
	Asthma	2 (4.2)	1 (2)	0.570
Medication, *n* (%)	ARB or ACEi	26 (54.2)	25 (53.2)	0.544
	BB	24 (50)	22 (46.8)	0.458
	CCB	13 (27.1)	10 (21.3)	0.337
	Statin	18 (37.5)	21 (44.7)	0.308

Values are presented as mean (range) for age, mean ± SD, or number (%). ASA-PS: American society of anesthesiology physical status, MI: Myocardiac infarction, HTN: Hypertension, DM: Diabetes mellitus, COPD: Chronic obstructive pulmonary disease, ARB: Angiotensin receptor blockers, ACEi: Angiotensin-converting enzyme inhibitors, BB: Beta blocker, and CCB: Calcium-channel-blocker.

**Table 2 ijerph-20-02310-t002:** Clinical characteristics.

	Group B (*n* = 48)	Group T (*n* = 47)	*p*-Value
Duration of surgery (min)	193.33 ± 28.48	187.45 ± 27.84	0.311
Duration of anesthesia (min)	244.06 ± 31.04	240.32 ± 32.74	0.569
Time of extubation (min)	883.75 ± 241.01	898.74 ± 256.50	0.770
Inotropes used (*n*)	45	44	0.651
Number of grafts	2.58 ± 0.58	2.45 ± 0.58	0.254
Amount of Sufentanil used (mcg)	295.82 ± 68.43	287.09 ± 70.51	0.542
Volume of fluid (ml/kg)	64.86 ± 21.65	67.85 ± 35.12	0.618
Number of RBC Transfusion (unit)	1.25 ± 1.16	1.36 ± 1.28	0.656
ICU length of stay (h)	26.56 ± 13.80	22.79 ± 8.40	0.112
Hospital length of stay (days)	13.79 ± 4.59	12.57 ± 3.33	0.143

Values are presented as mean ± SD. RBC: red blood cell and ICU: intensive care unit.

**Table 3 ijerph-20-02310-t003:** Perioperative QOR-40K score.

	Group B (*n* = 48)	Group T (*n* = 47)	*p*-Value
**Preoperative**			
Physical comfort	56.95 ± 3.92	55.91 ± 4.35	0.222
Emotional state	41.00 ± 4.37	39.47 ± 6.19	0.168
Psychological support	32.25 ±4.43	31.91 ± 4.53	0.512
Physical independence	22.48 ± 4.16	21.42 ± 4.75	0.253
Pain	33.85 ± 2.46	33.61 ± 2.06	0.612
Total score	186.54 ± 15.13	179.23 ± 26.29	0.102
**24 h after extubation**			
Physical comfort	50.75 ± 6.47	50.80± 5.19	0.961
Emotional state	37.48 ± 5.52	38.00 ± 5.54	0.647
Psychological support	29.79 ± 5325	30.74 ± 4.68	0.353
Physical independence	15.98 ± 5.90	15.13 ± 6.24	0.496
Pain	29.85 ± 3.36	30.13 ± 3.97	0.717
Total score	163.85 ± 15.32	164.80 ± 20.82	0.825
**48 h after extubation**			
Physical comfort	54.44 ± 3.93	53.27 ± 4.90	0.205
Emotional state	40.56 ± 3.96	39.23 ± 5.11	0.160
Psychological support	31.92 ± 4.18	30.87 ± 4.57	0.248
Physical independence	19.54 ± 4.41	19.36 ± 5.51	0.861
Pain	31.77 ± 2.83	31.74 ± 3.12	0.966
Total score	178.23 ± 15.32	174.49 ± 18.44	0.285

Values are presented as mean ± SD. QoR-40 = Quality of Recovery-40.

**Table 4 ijerph-20-02310-t004:** Post-operative pain.

	Group B	Group T	*p*-Value
NRS score at Extubation	5.54 ± 1.68	6.00 ± 1.68	0.186
Rescue analgesia, *n* (%) at extubation	19 (39.58)	22 (46.81)	0.477
NRS score at 24 h after extubation	4.60 ± 1.51	4.40 ± 1.65	0.540
Rescue analgesia, *n* (%) at 24 h after extubation	21 (43.75)	14 (29.79)	0.158
Fentanyl consumption at 24 h after extubation (mcg)	191.58 ± 98.75	189.41 ± 86.08	0.909
NRS score at 48 h after extubation	3.04 ± 1.15	3.06 ± 1.47	0.935
Rescue analgesia, *n* (%) at 48 h after extubation	6 (12.5)	6 (12.77)	0.969
Fentanyl consumption at 48 h after extubation (mcg)	368.56 ± 144.59	402.31 ± 145.85	0.260

Values are presented as mean ± SD or number (%). NRS: numeric rating scale.

**Table 5 ijerph-20-02310-t005:** Post-operative PONV.

	Group B	Group T	*p*-Value
PONV at extubation, *n* (%)	2 (4.17)	1 (2.13)	0.610
Rescue anti-emetics at extubation, *n* (%)	2 (4.17)	1 (2.13)	0.570
PONV at 24 h after extubation, *n* (%)	4 (8.33)	4 (8.51)	0.766
Rescue anti-emetics at 24 h after extubation, *n* (%)	2 (4.17)	4 (8.51)	0.384
PONV at 48 h after extubation, *n* (%)	3 (6.25)	3 (6.38)	0.979
Rescue anti-emetics at 48 h after extubation, *n* (%)	1 (2.08)	2 (4.26)	0.545

Values are presented as number (%). PONV: postoperative nausea and vomiting.

## Data Availability

The datasets generated and/or analyzed during the current study are not publicly available due to Hospital internal regulations but are available from the corresponding author on reasonable request.

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
