# Peer review of "Comparison of Postoperative Recovery between Balanced and Total Intravenous Anesthesia in Patients Undergoing Off-Pump Coronary Artery Bypass (OPCAB) Surgery: A Prospective, Single-Blind Randomized Study"

_ijerph, 2023, doi:10.3390/ijerph20032310_

Round 1

Reviewer 1 Report

Abstract: Need to clarify the randomized trial design; how was made participants allocation? Participants mentined but how much fulfilled the study requirements? Outcomes: what were the main outcomes measured?Randomization & Blinding: how were made? need to describe the process. Results: number randomized, recruitment, number analysed, outcome and harms poorly described. Trial registration and Funding not mentioned in the abstract.

Introduction: Background and objectives well described. Discussion of existing systematic review of the intervention. Knowledge gaps mentioned.

Methods: Participants eligibility criteria mentioned. Settings and locations where the data were collected are presumed but not declared. Central OR? Monovalent OR? A peripheral hospital or a major one? Interventions: please, provide the TIDieR checklist. Describe whether the intervention was taylored or modified. If intervention adherence or fidelity was assessed, describe the extent to which the intervention was delivered as planned. Outcomes: description of primary and secondary outcomes and when were assessed. There where any changes to trial outcomes after the trial commenced? Please, describe the reasons. Sample Size: declared. Randomization: please, clarify allocation concealment mechanisms (describe steps for allocation), implementation (who generate the allocation? who enrolled participants?), who was blinded?

Results: Participant flow well built. Recruitment, baseline data, outcomes estimation well described.

Discussion: the authors declare study limitations, and generalizability of the trial findings. Also, the interpretations are consistent with the results.

Other information: please, present where the full trial protocol can be access (if available).

Author Response

The authors thank reviewers for their enthusiastic and constructive criticism. The authors believe that these efforts will further advance the paper.

Comment

Abstract: Need to clarify the randomized trial design; how was made participants allocation? Participants mentined but how much fulfilled the study requirements? Outcomes: what were the main outcomes measured?Randomization & Blinding: how were made? need to describe the process. Results: number randomized, recruitment, number analysed, outcome and harms poorly described. Trial registration and Funding not mentioned in the abstract.

Response

Our abstract covered too limited content, and I agree with most of reviewer’s points. However, the number of words in an abstract is limited to 200 words according to the submission rules of this journal. We have tried to fix the pointed out within the word count limits, but we know it's not good enough.

Recovery after anesthesia has a significant impact on the patient’s return to daily life. This study was performed to compare postoperative quality of recovery according to the method of anesthesia in patients undergoing OPCAB using the Korean version of the Quality of Recovery-40 (QoR-40K) questionnaire. This single-blind prospective study (trial number: KCT0004726) was performed in a population of 102 patients undergoing OPCAB under general anesthesia. The patients were randomly assigned to one of two groups using a computer-generated list: a total intravenous anesthesia group (Group T) and a balanced anesthesia group (Group B). The QoR-40K score was measured preoperatively, and at 24 and 48 hours after extubation. There was no significant difference in the QoR-40K score between the groups at 24 and 48 hours after extubation. In addition, there were no significant differences between groups in any of the five dimensions of QoR-40K at 24 and 48 hours after extubation. Finally, there were no differences in the postoperative opioid consumption, time to extubation, or length of hospital stay. In this study, there was no difference in the QoR-40K score at 24 hours after extubation between Groups T and B. Therefore, both methods of anesthesia are suitable for use in OPCAB.

Comment

Introduction: Background and objectives well described. Discussion of existing systematic review of the intervention. Knowledge gaps mentioned.

Response

Thanks for your mention.

Comment

Methods:

Participants eligibility criteria mentioned. Settings and locations where the data were collected are presumed but not declared. Central OR? Monovalent OR? A peripheral hospital or a major one?

Response

As pointed out by the reviewer, we modified it as follows.

This prospective randomized single-blind controlled study was performed between March 2020 and August 2021 in single tertiary care hospital.

Comment

Interventions: please, provide the TIDieR checklist.

Response

As pointed out by the reviewer, we have attached the TIDieR checklist file.

Comment

 Describe whether the intervention was taylored or modified. If intervention adherence or fidelity was assessed, describe the extent to which the intervention was delivered as planned.

Response

Our study was not modified during the process, so there is nothing to further describe.

Comment

 Outcomes: description of primary and secondary outcomes and when were assessed.

Response

As the reviewer pointed out, it is important to establish a primary endpoint in the study, and it was poorly mentioned in our study. Therefore, the primary and secondary endpoints of our study were specifically described as follows.

The primary outcome was the QoR-40K score 24 h after extubation. Secondary outcomes were the QoR-40K score 48 h after extubation, peri-operative vital signs (MAP and HR), total amounts of sufentanil, fluid, and blood products administered during surgery, use of inotropics, time to extubation after surgery, length of stay in the ICU, and length of stay in the hospital. Pain score (NRS), use of rescue analgesics, post operative fentanyl consumption, post operative nausea and vomiting (PONV) and use of rescue antiemetics were also measured as a secondary outcome at the time of extubation and 24 and 48 h thereafter. All researchers in this study were anesthesiologists.

The patient’s HR and blood pressure, at the time of extubation and 24 and 48 h thereafter, as well as pain at the surgical site, the amount of fentanyl used in the pain control device, the use of additional analgesics, and the use of antiemetics were determined from the medical records. The numbers of transplanted vessels were confirmed based on the operation records. At 24 and 48 h after extubation, the questionnaires were completed by the participants themselves or by responding to what the researcher read to them.

Comment

There where any changes to trial outcomes after the trial commenced? Please, describe the reasons.

Response

There was no change in the study protocol after the start of the study.

Comment

Sample Size: declared.

Randomization: please, clarify allocation concealment mechanisms (describe steps for allocation), implementation (who generate the allocation? who enrolled participants?), who was blinded?

Response

In response to the reviewer's opinion. We have modified the following:

The group assignment process was performed based on a computer-generated list of random numbers by a researcher who did not participate in anesthesia. Researchers investigating the QOR-questionnaire did not know the assignment.

Comment

Results: Participant flow well built. Recruitment, baseline data, outcomes estimation well described.

Response

Thanks for your mention.

Comment

Discussion: the authors declare study limitations, and generalizability of the trial findings. Also, the interpretations are consistent with the results.

Response

Thanks for your mention.

Comment

Other information: please, present where the full trial protocol can be access (if available).

Response

We added to attachments file

Reviewer 2 Report

The manuscript describes patient satisfaction after OPCAB surgery. Quality of recovery was assessed by QoR-40K score, postoperative pain and PONV. The authors hypothesized that the degree of recovery is different between intravenous and inhalation anesthesia. The manuscript is well written. The results answer the research question.

My comments:

1.       Why do the authors use the term “balanced anesthesia”?

2.       The authors do not present other methods of quality of recovery after anesthesia.

3.       The maintenance of anesthesia stage in groups T and B is not described clearly enough in Methods.

4.       Lines 63 to 79 deal with advantages of inhalation and intravenous anesthesia in cardiac surgery. This is not the topic of this paper and should be removed. Especially since this topic has been researched and presented in numerous publications.

5.       Also for this reason, lines 90-93 in Conclusion do not add anything new to this topic.

6.       In Abstract, the abbreviation OPCAB was used for the first time, without the full name. The full name is in the title, but there is no abbreviation.

7.       In table 2, the full name of the abbreviation “RBC” is missing.

8.       In Tables 4 and 5, the line “rescue analgesia” should be extended with after 24h and after 48h. The abbreviation NRS is not described in Table 4.

Author Response

The authors thank reviewers for their enthusiastic and constructive criticism. The authors believe that these efforts will further advance the paper.

Comment 1.

Why do the authors use the term “balanced anesthesia”?

Response 1.

Balanced anesthesia is the counterpart to intravenous anesthesia and is a term often used in general anesthesia. In fact, it is also widely used in other studies. As an example, some studies cited more than 50 times are introduced below. Therefore, our authors chose the words intravenous anesthesia and balanced anesthesia. However, if the reviewer thinks that inhalation anesthesia can better convey the meaning of our study, we are willing to change it.

HOHLRIEDER, M., et al. Effect of total intravenous anaesthesia and balanced anaesthesia on the frequency of coughing during emergence from the anaesthesia. British journal of anaesthesia, 2007, 99.4: 587-591.

LEDOWSKI, Thomas, et al. Neuroendocrine stress response and heart rate variability: a comparison of total intravenous versus balanced anesthesia. Anesthesia & Analgesia, 2005, 101.6: 1700-1705.

SLOAN, Tod B., et al. Intraoperative neurophysiological monitoring during spine surgery with total intravenous anesthesia or balanced anesthesia with 3% desflurane. Journal of clinical monitoring and computing, 2015, 29.1: 77-85.

Comment 2.

The authors do not present other methods of quality of recovery after anesthesia.

Response 2.

There are few questionnaires about postoperative recovery that have been translated into Korean and validated. In addition to Qol-40k, there is QOL-15K, but QoL-15k was not used because it was derived from QoL-40k.

As pointed out by the reviewer, we modified it as follows.

These include numerical score, postoperative quality recovery score (PQRS), and quality of postoperative recovery (QOR).

Comment 3.

The maintenance of anesthesia stage in groups T and B is not described clearly enough in Methods.

Response 3.

It was added that anesthesia was maintained with a target BIS(Bisepectral insex) of 40-60.

Comment 4.

Lines 63 to 79 deal with advantages of inhalation and intravenous anesthesia in cardiac surgery. This is not the topic of this paper and should be removed. Especially since this topic has been researched and presented in numerous publications.

Response 4.

We thank the reviewer's reasonable and appropriate suggestion. In response to the reviewer's opinion, the content was deleted.

Comment 5.

Also for this reason, lines 90-93 in Conclusion do not add anything new to this topic.

Response 5.

In response to the reviewer's opinion, the content was deleted.

Comment 6.

In Abstract, the abbreviation OPCAB was used for the first time, without the full name. The full name is in the title, but there is no abbreviation.

Response 6.

Thank for your comment, we corrected it.

Comment 7.

In table 2, the full name of the abbreviation “RBC” is missing.

Response 7. 

Thank for your comment, we corrected it.

Comment 8.

In Tables 4 and 5, the line “rescue analgesia” should be extended with after 24h and after 48h. The abbreviation NRS is not described in Table 4.

Response 8.

Thank for your comment, we corrected it.